# *Mycoplasma agalactiae*: The Sole Cause of Classical Contagious Agalactia?

**DOI:** 10.3390/ani11061782

**Published:** 2021-06-15

**Authors:** Sergio Migliore, Roberto Puleio, Robin A. J. Nicholas, Guido R. Loria

**Affiliations:** 1Istituto Zooprofilattico Sperimentale della Sicilia “A. Mirri”, 90129 Palermo, Sicily, Italy; sergio.migliore@izssicilia.it (S.M.); guidoruggero.loria@izssicilia.it (G.R.L.); 2Consultant, The Oaks, Nutshell Lane, Farnham, Surrey GU9 0HG, UK; robin.a.j.nicholas@gmail.com

**Keywords:** contagious agalactia, mycoplasmas, *Mycoplasma agalactiae*, etiology, small ruminants

## Abstract

**Simple Summary:**

For over thirty years, contagious agalactia has been recognized as a mycoplasma disease affecting small ruminants caused by four different pathogens: *Mycoplasma agalactiae*, *Mycoplasma mycoides* subsp. *capri*, *Mycoplasma capricolum* subsp. *capricolum* and *Mycoplasma putrefaciens* which were previously thought to produce clinically similar diseases. Today, with major advances in diagnosis enabling the rapid identification by molecular methods of causative mycoplasmas from infected flocks, it is time to revisit this issue. In this paper, we discuss and argue the reasons to support *Mycoplasma agalactiae* infection as the sole cause of contagious agalactia.

**Abstract:**

Contagious agalactia (CA) is suspected when small ruminants show all or several of the following clinical signs: mastitis, arthritis, keratoconjunctivitis and occasionally abortion. It is confirmed following mycoplasma isolation or detection. The historical and major cause is *Mycoplasma agalactiae* which was first isolated from sheep in 1923. Over the last thirty years, three other mycoplasmas (*Mycoplasma mycoides* subsp. *capri*, *Mycoplasma capricolum* subsp. *capricolum* and *Mycoplasma putrefaciens*) have been added to the etiology of CA because they can occasionally cause clinically similar outcomes though nearly always in goats. However, only *M. agalactiae* is subject to animal disease regulations nationally and internationally. Consequently, it makes little sense to list mycoplasmas other than *M. agalactiae* as causes of the OIE-listed CA when they are not officially reported by the veterinary authorities and unlikely to be so in the future. Indeed, encouraging countries just to report *M. agalactiae* may bring about a better understanding of the importance of CA. In conclusion, we recommend that CA should only be diagnosed and confirmed when *M. agalactiae* is detected either by isolation or molecular methods, and that the other three mycoplasmas be removed from the OIE *Manual of Diagnostic Tests and Vaccines in Terrestrial Animals* and associated sources.

## 1. Introduction

In 1999, the consensus of the working group on contagious agalactia (CA) of the EC COST Action 826 on ruminant mycoplasmoses, which met in Toulouse, France, agreed that four mycoplasmas—*Mycoplasma agalactiae*, *M. mycoides* subsp. *capri* (previously named *M. m. mycoides* large colony), *M. capricolum* subsp. *capricolum* and *M. putrefaciens*—should be recognized as causal agents of CA because the clinical disease they cause can be similar and includes mastitis, arthritis, keratoconjunctivitis and, occasionally, abortion [1]. This decision followed earlier observations by Perreau in 1979 [2] that clinical signs of infections causing *M. agalactiae*, the main and historical cause, were sufficiently similar to those of *M. m. capri* and *M. c. capricolum* to be considered CA. As a result, these were listed in the OIE Manual of Diagnostic Tests and Vaccines in Terrestrial Animals in 1988 [3]. Furthermore, Lambert, who compiled the chapter on CA, added another mycoplasma, *M. putrefaciens*, mainly based on a few reports including one from Da Massa et al. in 1987 describing a severe outbreak of mastitis and arthritis in goats requiring the slaughter of nearly 700 goats [4]. Despite the infrequency of reported cases of *M. putrefaciens* and, indeed, *M. c. capricolum*, the four mycoplasmas have continued to be listed in their manual by the OIE ever since as etiological agents of CA. The inclusion of all four mycoplasmas has not been without argument with the OIE Collaborating Centre for International Cooperation in Animal Biologics at Iowa State University commenting in 2004: *“Some authorities consider infections with all of these agents to be CA; others prefer to reserve the term for infections with (only) M. agalactiae. Until this issue has been resolved, reports of CA outbreaks should specify the species involved”* [5]. Unfortunately, few authorities have followed this advice even in the OIE’s Terrestrial Animal Health Code, World Animal Health publications or their online equivalents.

Today, with major advances in diagnosis enabling the rapid identification by molecular methods of causative mycoplasmas from infected small ruminants, which is required to confirm diagnosis of CA, it is belatedly time to revisit this issue. This is important because only *M. agalactiae* is recognized nationally and internationally and subject to animal disease regulations covering CA. While *M. m. capri* is a widespread and probably underestimated respiratory pathogen, the disease it causes is, in most cases, clinically distinct from that caused by *M. agalactiae*. *M. c. capricolum* also favors the respiratory system and is rarely reported, while *M. putrefaciens* can cause reduction in milk production but affected animals are often without clinical signs. Moreover, *M. m. capri*, *M. c. capricolum* and *M. putrefaciens* are pathogens of goats rarely affecting sheep, which are by far the most economically important small ruminant species worldwide, particularly in the EU, where sheep numbers are seven times higher than those of goats [6]. For these reasons, the principal objective of this paper is to show that *M. agalactiae* should be considered the sole cause of classical CA.

## 2. General Consideration about Contagious Agalactia

### 2.1. Clinical Findings and Epidemiology

The main clinical signs of CA caused by *M. agalactiae* are mastitis which can involve 60–80% of lactating females, followed by arthritis, keratoconjunctivitis and abortion in less than 10% of affected animals. Clinical signs can be seen at various stages during the evolution of the disease, not necessarily in the same animal but in individuals in the affected flock or herd [7]. *M. agalactiae* accounts for 90% of outbreaks of CA in goats [8] and almost 100% in sheep [7]; it is characterized by high morbidity (sometimes up to 50 and 90% of the lactating female sheep and goats, respectively), drastic reduction of milk production and, in 90% of cases, the most visible sign, interstitial mastitis, is seen [9]. Arthritis and keratoconjunctivitis are normally observed in 5–10% of cases [10]. Respiratory disease is very rarely a feature of CA caused by *M. agalactiae* and, where it has been reported, may have been the result of a mixed infection with other pathogens, in particular *M. m. capri* [11].

The three other mycoplasmas which have been added to the etiology of CA can occasionally cause clinically similar outcomes, though exclusively in goats. However, the two pathogens belonging to the *M. mycoides* group, *M. m. capri* and *M. c. capricolum*, are more often isolated from pneumonic goats [12] or from polyarthritic kids [13] and only rarely reported in sheep in some areas [14].

In an effort to separate diseases caused by *M. m. capri* and other pulmonary mycoplasmas in goats from CA, Thiaucourt and Bolske used the rather awkward term MAKePs syndrome standing for mastitis, arthritis, keratoconjunctivitis and pneumonia [15]. Outbreaks often show high morbidity of up to 40, 80 and 90% in adults, lactating females and kids respectively; mastitis, on the other hand, is rarely reported compared to the other more predominant signs such as severe respiratory or poly-arthritic syndromes and, even less frequently, conjunctivitis, abortions and stillbirth [7].

The fourth listed mycoplasma *M. putrefaciens* is a very infrequent isolate from goats with questionable pathogenicity as it is often isolated from healthy goats but has been found occasionally in goat herds presenting agalactia. It is often found with *M. agalactiae* where it may play little role in disease progression [16,17]. However, De Massa et al. [4], reported a serious outbreak in the USA requiring the destruction of 700 goats. However, the authors concluded that poor hygiene involving infusion of the pathogen into the teat canal and the feeding of raw colostrum played a major part in the disease. Its pathogenicity seems primarily to depend on intrinsic and external hosts factors [7].

Overall, most countries involved in small ruminant dairy production identify *M. agalactiae* as the unique or major pathogen in CA. Some European countries also report to national authorities some or all of the three mycoplasmas, which may not always be linked to outbreaks of mastitis and/or falls in milk production. An investigation carried out between 2004 and 2012 in Sardinia, which has the highest sheep population in Italy, found a high number of outbreaks of *M. m. capri.* This was shown to be the result of introductions of goat breeds from Spain and other European countries. The disease was confirmed in 34 goat farms and a single sheep flock [18]. While the authors did not record the clinical signs of the individual herds, it is interesting to note that *M. m. capri* was isolated from the lungs and brains of over 74% of the herds, sites rarely associated with *M. agalactiae* (7).

Clinical signs of the four mycoplasmas are summarized in Table 1.

### 2.2. Pathological Findings in Experimental and Natural Infection

It can be difficult to make comparisons between infections caused by these various mycoplasmas because experimental infections do not always mimic natural infection as the pathogen may invade less aggressively either following contamination from the environment or the infected hands of the milkers. However, reports have shown quite distinct pathological changes following infection with different mycoplasmas. Moreover, lesions resulting from experimental infections of *M. agalactiae* vary according to the route of infection. Hasso and colleagues [19] reported an experimental infection with *M. agalactiae* testing four different routes of inoculation and observed: acute to chronic mastitis in the intramammary and subcutaneously inoculated groups; acute synovitis in the intravenously inoculated group and, to a lesser extent, in the intramammary inoculated group; and subacute enteritis in the orally inoculated group. No changes were detected in the eyes.

Other authors [20,21] have reported pathology induced by *M. m. capri* infection as a disease which involves mainly the thoracic cavity in adults and the joints of young animals. There was no mention of lesions in the mammary glands. Studies carried out in Italy some years ago [22] confirmed that *M. m. capri* infection was a respiratory and poly-arthritic syndrome and only one of ten experimentally infected goats had udder lesions detected by immunohistochemistry. Agnello et al. [15] investigated a natural outbreak in local goats caused by *M. m. capri* where few adult animals showed clinical signs or lesions, whereas a severe poly-arthritic syndrome was found in the 90% of kids with an 80% mortality rate.

Bergonier et al. [7] investigated the anatomic location of *M. agalactiae* and *M. m. capri* in goats following natural infection and showed that *M. m. capri* possessed a greater respiratory tropism than *M. agalactiae*. In addition, a large number of *M. m. capri* isolates were found in the ear canal, demonstrating a close connection between the respiratory system and the middle ear.

Pathological and immunohistochemical findings observed in 12 kids experimentally infected with *M. c. capricolum*, *M. m. capri* and *M. m. mycoides* (large colony type) showed fatal septicemia within 5 days; histopathological findings consisted of acute diffuse interstitial pneumonia, arthritis and multifocal necrotic purulent splenitis [23].

The few studies on *M. putrefaciens* pathology have shown that only intramammary inoculation can lead to an acute mastitis while other routes failed to produce lesions [24]; this may suggest the opportunistic nature of this mycoplasma probably overestimated for its pathogenic role. Experimental infection with *M. putrefaciens* strains in lactating goats via intramammary inoculation caused an increase of leukocytes in milk within 8 days but no sign of mastitis in spite of a complete halt in lactation after some days [25]. Pathological findings are summarized in Table 1.

Da Massa [24] reported as “cardinal lesions” of infections with *M. c. capricolum*, a fibrinopurulent polyarthritis and an acute, diffuse interstitial pneumonia. However, lactating goats exposed to low numbers of the organism via the teat canal experienced similar lesions as well as acute mastitis, agalactia and hardened udders.

### 2.3. Geographical Location

Of potential significance to our argument is the geographical distribution of the mycoplasmas and identification of regions where CA is reported. Unfortunately, this is difficult to be certain of as reports of CA by the World Animal Health Information System (WAHIS) issued by the OIE do not specify the pathogen involved. Furthermore, many reporting countries do not have the ability or inclination to identify mycoplasmas, so diagnosis is mainly based on clinical disease. In countries where mycoplasma laboratories exist, CA is only declared when *M. agalactiae* is isolated. Indeed, few, if any, countries report CA officially when *M. m. capri*, *M. c. capricolum* and *M. putrefaciens* are detected. Figure 1 shows the countries reporting CA in 2019, although some report that the disease is restricted to certain zones. These are surprisingly few and include, mostly, countries surrounding the Mediterranean and several in Western Asia and South America. The USA and Canada suspect the disease but there have been only sporadic reports of isolations of *M. agalactiae* and very few reports of the other causative mycoplasmas over the last 20 years [26]. Many other countries have reported isolations of these mycoplasmas but it seems regular monitoring does not take place. Jordan appears in the WAHIS “suspect” category but has reported most of these mycoplasmas several times [27]. In Bosnia and Herzegovina, cases of *M. m. capri*, *M. c. capricolum* and *M. putrefaciens* in both sheep and goats have been reported [28]. In North Macedonia [29] and in Turkey [30], *M. agalactiae* appears to be dominant in both sheep and goats. In North and Central Spain, *M. agalactiae* is widely present in sheep but *M. m. capri*, *M. c. capricolum* and *M. putrefaciens* remain unreported [31]. In Murcia, a mycoplasma survey of goats reported a high prevalence of CA (67%) dominated by *M. agalactiae* (75%) compared to *M. m. capri* (4%) [32]. In contrast, in the Canary Islands, the prevalence of *M. m. capri* in goats is similar to that of *M. agalactiae* [33].

In France, a national surveillance network estimated a prevalence of 42%, 26% and 15% for *M. m. capri*, *M. c. capricolum* and *M. putrefaciens* respectively but rarely detected *M. agalactiae* in clinically affected goats [34]. In Italy, *M. agalactiae* is considered as a dominant pathogen in major sheep-breeding regions [12], whereas *M. m. capri* strains were also isolated in goats in Sardinia and Sicily [15]. In addition, *M. c. capricolum* and *M. putrefaciens* were isolated from a few milk samples in Sardinia in 2018 (Tola, personal communication). In Greece and Cyprus, all reports in both sheep and goats have focused exclusively on *M. agalactiae* [35].

More recently, CA has been reported in countries in Western Asia like Iran and Mongolia where automation of the dairy industry is still rare. While isolation of *M. m. capri* has been reported in Australia and New Zealand, CA has never been diagnosed despite the large numbers of sheep probably because most are maintained for meat production. In general, *M. agalactiae* appears to have a more restricted distribution: Southern Europe, West Asia, including Turkey and Iran, and North Africa with occasional reports from parts of South America like Brazil [36]. On the other hand, *M. m. capri* has been reported on most continents of the world where small ruminants are kept. Reports of *M. c. capricolum* and *M. putrefaciens* are much more sporadic and infrequent.

### 2.4. Characteristics of the Causative Mycoplasmas

It is worth noting that *M. agalactiae* is very closely related to the bovine pathogen, *M. bovis*, and grouped in the *hyorhinis* group along with other animal pathogens. The other three mycoplasmas, *M. m. capri*, *M. c. capricolum* and *M. putrefaciens*, belong or are very closely related to the genetically distant *mycoides* cluster surprisingly located in the spiroplasma group, *mollicutes* more commonly found in plants and insects. Indeed, *M. agalactiae* shares only 18% of its genome with the *M. mycoides* cluster [37]. Interestingly, data reports on antibiotic resistance profiles of CA-causing *Mycoplasma* spp. showed a marked difference in behavior in vitro. Erythromycin is effective against infection by *M. m. capri*, *M. c. capricolum* and *M. putrefaciens*, but inefficient against *M. agalactiae* strains [12,14,27]. This is an additional confirmation that there are two different kinds of diseases related to whether the pathogen is a member of the *bovis* or *mycoides* groups.

### 2.5. Legislation

In the United Kingdom, like many other disease-free countries, CA is a notifiable disease and is covered by two pieces of legislation: the Specific Diseases (Notification and Slaughter) Order 1992 and the Specific Diseases (Notification) Order 1996 which describe procedures to be taken, including movement restriction, slaughter and disinfection, in the event of an outbreak. However, action is only taken if *M. agalactiae* is suspected as was a recent case involving imported goats from France [38]. Poland also has specific legislation for CA but only enforceable if *M. agalactiae* is detected. Israel regularly isolates *M. agalactiae* and *M. m. mycoides* but only reports this nationally while informing the OIE that CA is present based on clinical findings. In many countries including Turkey and Portugal, cases of CA are not reported possibly because of the complexities of reporting several pathogens as causes of CA. In France, Greece, Iran and Italy as well, action is only taken in the event of the detection of *M. agalactiae*. Indeed, we do not know of any country that reports or takes action if any of the other causative mycoplasmas are found.

## 3. Discussion

In countries where *M. agalactiae* represents the most important and prevalent pathogen associated with CA, the disease shows the same clinical course both in sheep and goats. Moreover, in those countries like Italy where often both animal species are kept together in the same group, owners do not report any clinical differences between them. It seems to confirm that the clinical behavior of CA is quite different according to the pathogen involved, target species, breed, farm/husbandry type or degree of domestication. However, from a veterinary point of view, two different diseases are seen. First, the “typical or classical disease” which is very contagious affecting flock milk production and for this reason is called “contagious agalactia”. This disease occurs exclusively in milking/dairy small ruminants. In the ancestral species, the Spanish ibex (*Capra pirenaica*), the clinical signs are represented by blindness, malnutrition and polyarthritis [39], but there are no reports of mastitis probably because this lesion is mainly related to the last phase of species domestication and its dairy purpose. The second clinical syndrome related to CA is linked to the other mycoplasma infections and generally causes other clinical signs probably initiating in the respiratory tract of adult goats and joints in kids; only a small percentage of infections may involve the udder.

The introduction of European Union (EU) regulation 2016/429, which provided a new regulatory framework covering control and surveillance of infectious diseases, animal welfare and animal movements amongst Member States, complements local legislation but also reduces the amount of intervention possible [40]. Worryingly, this directive will impact some traditionally notifiable diseases, including CA, and remove them from the new EU disease-listing process, with control being devolved to the national or even local level. This apparent downgrading of the importance of CA may negatively affect international collaboration on control across borders and make this disease a less likely area for research funding. Delisting of CA from EU notifiable diseases [41] associated with the absence of surveillance plans and of any other mandatory certification for trades may also facilitate the risk of introduction of CA “healthy carriers” in CA-free small ruminant populations which may enable a massive spreading of the disease. Conversely, the identification of *M. agalactiae* as a single cause of CA may help focus attention on this important pathogen and remove the confusion and uncertainty that exists about reporting of this OIE-listed disease. This will also fit better into the OIE ethos of “one disease, one cause”. It is hoped that this paper may cause the EU to reconsider altering the status of CA in future amendments to the regulation as we believe it is an under-reported disease with international significance.

Finally, in a further effort to bring about clarity to this complex area we propose a new term for the disease caused by the largely respiratory pathogens *M. m. capri* and *M. c. capricolum*: caprine respiratory and articular syndrome (CRAS). This reflects more accurately the main disease signs and separates it from the OIE-listed contagious caprine pleuropneumonia caused by the closely related *M. capricolum* subsp. *capripneumoniae* which is confined to the thoracic cavity.

## 4. Conclusions

In conclusion, we recommend that CA should only be diagnosed and confirmed when *M. agalactiae* is detected either by isolation or molecular methods such as PCR and the other three mycoplasmas removed from the OIE *Manual of Diagnostic Tests and Vaccines in Terrestrial Animals* and associated sources.

## Figures and Tables

**Figure 1 animals-11-01782-f001:**
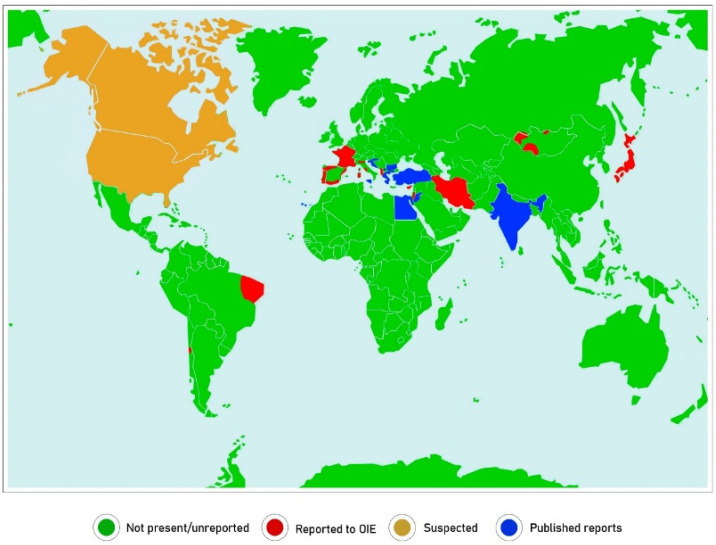
Countries reporting the mycoplasmas causing contagious agalactia worldwide.

**Table 1 animals-11-01782-t001:** Features of the contagious agalactia (CA) pathogens group.

PATHOGENS		*M. agalactiae*	*M. m. capri* *M. c. capricolum* *M. putrefaciens **
GENE CLUSTER		*M. bovis*	*M. mycoides*
HOST SPECIES		Sheep; goats	Goats; sheep (±)
CLINICAL FORM	sheep	subacute/chronic	Not reported
	goats	acute/chronic	hyperacute/subacute (Mmc, Mcc)
MORTALITY	sheep	low	Not reported
	goats	low	High (especially in kids)
MORBIDITY	sheep	>50%	Not reported
	goats	>70%	>80%
TYPICAL SYNDROMES		Mammary Articular Ocular Others	Pulmonary Articular Ocular Others
PATHOLOGY	Frequent (>60% of cases)	Chronic interstitial mastitis (Unilateral or bilateral)	- Primary interstitial pneumonia and pleurisy; - Severe fibrin purulent polyarthritis; - Septicemia (kids)
	Rare (<15% of cases)	- Keratoconjunctivitis; - Fibrinopurulent arthritis; - Pneumonia and pleurisy; - Abortion	- Chronic interstitial mastitis (unilateral or bilateral); - Keratoconjunctivitis; - Abortion

* Very few clinical and pathological reports.

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
