# Peer review of "Mycoplasma agalactiae: The Sole Cause of Classical Contagious Agalactia?"

_animals, 2021, doi:10.3390/ani11061782_

Round 1

Reviewer 1 Report

Mycoplasma agalactiae: the sole cause of classical contagious agalactia.

General comments:

Admittedly, I have never encountered a case of CA in small ruminants, nor am I familiar with the EU regulations in this area.  I did find the article an interesting read, and one that is worthy of publication because it appears to be an area that requires further debate.

That said, I think the article could undergo some major editing.

I think the objective of the paper needs to be stated more clearly.  To me, it seems as though the authors should be advancing an argument that CA in sheep is distinct from CA of small ruminants, and hence could be addressed by renaming it Ovine contagious agalactiae or something to this effect?  I think paragraph 62-72 needs to be rewritten because I personally found it difficult to follow.  It starts with stating that we can now more accurately identify the etiological agents of CA.  Next, M. m capri is an underestimated problem, but M. agal is only regulated pathogen -not sure how these two points connect. Then, M. c capri and M putre are rarely found and never reported, and the three non M. agal are pathogens of goats, but perhaps of economical insignificance. Concluding with M. agal should be the sole cause of CA.  I think if “sheep” was added at the end of the sentence, it would make more sense. 

My sense is that lines 71-72 is advocating for ignoring M mycoides gene cluster species because they don’t occur in sheep. Again, I think this is a name change to the disease, making it more narrowly defined and descriptive.  My concern is that is sounds as though the authors are advocating for not surveilling for the M. mycoides organisms, at least in sheep. The fact that we can now distinguish the etiologies would suggest that we should indeed continue to monitor and report on these diseases, since they could conceivably become emerging dz of sheep.

Lastly, the EU regulatory changes in the Discussion seem to be signficant, and worthy of more discussion. Particularly, placing these changes within the context of the article. Why are these changes occurring? I like the last line “one disease, one cause” ethos.  Hence Ovine CA would separate this from the polymicrobial nature of CA in all small ruminants. Akin to stating bovine pastuereollosis vs bovine respiratory disease. 

Specific comments – some repeat what was stated above.

 The subtitles don’t accurately describe the narrative to follow.

- Lines 17-18 -  no need to say all, if a dx can be made if several is sufficient.  However, is this correct, can you definitely dx CA based on presence of mastitis and abortion alone, for example.

- Lines 18 and 44.  Line 22-23 states M. mycoides subsp capri is a major cause of pneumonia, but is then not listed as a C/S for CA in small ruminants. 

- Line 72 – perhaps this should be modified as to be considered the sole cause of CA in sheep.  Thus, CA is a catch-all phrase for many different pathogens, similar to Bovine Respiratory Disease being used to describe all pneumonias in cattle, regardless of etiology. 

- 2.1 Clinical findings – perhaps this is should be titled Clinical findings and Epidemiology. Perhaps Table 1 should be divided into two, with the top half of the table being inserted into the secition on clinical findings and epidemiology.  You could then remove a significant amount of the narrative.

- 75-76 – why repeat this line.

- Line 76-77 – I wonder if the bigger issue is the name CA, since this specifically describes agalactiae (mastitis), but the disease manifests with other C/S.  How prevalent is mastitis compared to the other clinical outcomes?

- Lines 87-91.  This needs to be broken into at least two sentences.  Confusing as is written. Not sure what stat significance is referring too – prevalence or Incidence of C/S?

- 2.2 Pathological findings – this section seems more directed to explaining experimental challenges than describing gross and histopathological findings. This is a bit problematic since the first line states that experimental infections does not mimic natural infections.

Table 1 – can you futher define ‘rare’ and ‘frequent’.

2.3 Geographical location – I would delete distribution from Table 1, since it is adequately covered in this section. Would a map showing the distribution of the 4 pathogens be more informative than desribing it in the narrative?

Line 214-215:  I think the description of erythromycin suscept/resistance in this section is sufficient that it does not need to be incldued in Table 1.  This doesn’t add much to the table and no point in repeating it twice. Same for sugars catabolism. 

Lines 233-248 Discussion – opening paragraph is very good, but it would have been helpful to have read this in the Introduction or perhaps when discussion the clin finding and epi.

Lines 249-263.  This concern regarding reporting and the delising of CA as a Notifiable Disease comes as a surprise at the end of the article. The Objectives of the manscript in Lines 62-72 makes no reference to changes in regulatory status and why this is a concern. Rather, the premise of the article is that CA should be more tightly defined as a disease, primarily of sheep, caused by M. agal.

Author Response

Response to reviewer 1

Rev- Mycoplasma agalactiae: the sole cause of classical contagious agalactia. General comments: Admittedly, I have never encountered a case of CA in small ruminants, nor am I familiar with the EU regulations in this area. I did find the article an interesting read, and one that is worthy of publication because it appears to be an area that requires further debate. That said, I think the article could undergo some major editing. I think the objective of the paper needs to be stated more clearly.

We have tried to clarify what are quite difficult issues and hopefully make the objective clearer. The paper has undergone major editing particularly section 2.1

To me, it seems as though the authors should be advancing an argument that CA in sheep is distinct from CA of small ruminants, and hence could be addressed by renaming it Ovine contagious agalactiae or something to this effect?

Not exactly, our goal is the association "one disease, one cause" and propose the term contagious agalactia exclusively for M. agalactiae infections in both sheep and goats, in order to distinguish m. agalactiae infection from the other three pathogens, causative agents of a syndrome whose name should be reassigned. We have tried to explain this concept better in the text.

I think paragraph 62-72 needs to be rewritten because I personally found it difficult to follow. It starts with stating that we can now more accurately identify the etiological agents of CA. Next, M. m capri is an underestimated problem, but M. agal is only regulated pathogen -not sure how these two points connect. Then, M. c capri and M putre are rarely found and never reported, and the three non M. agal are pathogens of goats, but perhaps of economical insignificance.

This paragraph has been completely revised  L63-78

Concluding with M. agal should be the sole cause of CA. I think if “sheep” was added at the end of the sentence, it would make more sense.

In our opinion, it would be wrong to declare this that the disease is only related to sheep as goats suffer too

My sense is that lines 71-72 is advocating for ignoring M mycoides gene cluster species because they don’t occur in sheep. Again, I think this is a name change to the disease, making it more narrowly defined and descriptive. My concern is that is sounds as though the authors are advocating for not surveilling for the M. mycoides organisms, at least in sheep.

Perhaps we have given the wrong impression but the reason we believe that these mycoplasmas cause in the vast majority of cases a clinically distinct disease from CA and the decision to include them as aetiological agents was misguided. Perhaps we also gave the wrong impression concerning surveillance of the cluster members. In a perfect world this of course should be done and indeed is being done by some countries but these mycoplasmas are not covered by regulations so there is no compulsion to look for them. We hope the next text reflects this  L67-69

The fact that we can now distinguish the aetiologies would suggest that we should indeed continue to monitor and report on these diseases, since they could conceivably become emerging dz of sheep. Lastly, the EU regulatory changes in the Discussion seem to be significant, and worthy of more discussion.

Absolutely! As said above this is done at the national level and often published but there is no commitment to do this at international level or likely to happen in the near future. We have added some additional discussion on the EU changes L276-277

Particularly, placing these changes within the context of the article. Why are these changes occurring? I like the last line “one disease, one cause” ethos. Hence Ovine CA would separate this from the polymicrobial nature of CA in all small ruminants. Akin to stating bovine pasteurellosis vs bovine respiratory disease.

Goats present a clinically identical if more serious disease than sheep when infected with M agalactiae so it would not in our opinion be a sensible decision to exclude goats so it really will be a case of : “one disease, one cause, several animals” which is in line with the OIE

Specific comments – some repeat what was stated above. The subtitles don’t accurately describe the narrative to follow. - Lines 17-18 - no need to say all, if a dx can be made if several is sufficient. However, is this correct, can you definitely dx CA based on presence of mastitis and abortion alone, for example.

Subtitles have been changed. We have altered the text to make it clearer that diagnosis is only confirmed as CA following lab tests L20, 65-66. I think it is necessary to keep “all” as this would be the ideal scenario for suspecting CA

- Lines 18 and 44. Line 22-23 states M. mycoides subsp capri is a major cause of pneumonia, but is then not listed as a C/S for CA in small ruminants. 2 - Line 72 – perhaps this should be modified as to be considered the sole cause of CA in sheep. Thus, CA is a catch-all phrase for many different pathogens, similar to Bovine Respiratory Disease being used to describe all pneumonias in cattle, regardless of aetiology.

We have changed the text to clarify our argument L63-78

- 2.1 Clinical findings – perhaps this  should be titled Clinical findings and Epidemiology. Perhaps Table 1 should be divided into two, with the top half of the table being inserted into the section on clinical findings and epidemiology. You could then remove a significant amount of the narrative.

We have removed several items from the table but would like to keep a single table

- 75-76 – why repeat this line.

The sentence was removed.

 - Line 76-77 – I wonder if the bigger issue is the name CA, since this specifically describes agalactiae (mastitis), but the disease manifests with other C/S. How prevalent is mastitis compared to the other clinical outcomes?

Amended. We have also suggested an alternative name for the largely pulmonary pathogens as recommended by ref 2

 - Lines 87-91. This needs to be broken into at least two sentences. Confusing as is written. Not sure what stat significance is referring too – prevalence or Incidence of C/S?

Sentence(s) altered L99-101

 - 2.2 Pathological findings – this section seems more directed to explaining experimental challenges than describing gross and histopathological findings. This is a bit problematic since the first line states that experimental infections does not mimic natural infections.

The subtitle was changed to “Pathological findings in experimental and natural infection”. Perhaps we overstated the difficulty of mimicking natural from experimental infection but we think it is a valid method to show further evidence that the cluster members produce a clinically distinct disease in goats. We have altered the text accordingly.  L128-129

Table 1 – can you further define ‘rare’ and ‘frequent’.

OK done, see the text 

2.3 Geographical location – I would delete distribution from Table 1, since it is adequately covered in this section. Would a map showing the distribution of the 4 pathogens be more informative than describing it in the narrative?

We have deleted distribution from table. This is a good suggestion, but unfortunately, the OIE reports only Ma disease and doesn’t discriminate all  CA pathogens involved in the outbreak.

Line 214-215: I think the description of erythromycin suscept/resistance in this section is sufficient that it does not need to be included in Table 1. This doesn’t add much to the table and no point in repeating it twice. Same for sugars catabolism.

Table 1 was modified

Lines 233-248 Discussion – opening paragraph is very good, but it would have been helpful to have read this in the Introduction or perhaps when discussion the clin finding and epi.

Thank you for that suggestion but we would prefer to leave it there as it sums up our results and widens the discussion somewhat

Lines 249-263. This concern regarding reporting and the delisting of CA as a Notifiable Disease comes as a surprise at the end of the article. The Objectives of the manuscript in Lines 62-72 makes no reference to changes in regulatory status and why this is a concern. Rather, the premise of the article is that CA should be more tightly defined as a disease, primarily of sheep, caused by M. agal.

These changes are fairly new and we have written a separate paper on our concerns but there are opportunities that having a single agent as a cause of CA may remove confusion about this disease. However, the main target of our paper is the OIE who have no plans that we are aware of to remove CA from their list of diseases. As a result, we hope they will reconsider the multi-aetiology of CA presently contained in the Manual and elsewhere L282-285

Reviewer 2 Report

The authors promote a worthwhile discussion regarding the aetiology of contagious agalactia. Evidences such as clinical presentation, pathogenesis and geographic distribution are discussed in an appropriate depth. The text organization is logical . However, the article would benefit from more work in connecting ideas at sentence level. Sentences are often too long which hampers the cohesion of arguments presented.  As this scientific paper style strongly relies on an effective argumentation process, careful check and proofread of the manuscript is strongly recommended.

In addition, the fact that Mycoplasma mycoides subsp. Capri appears to have a different pathogenesis, as pulmonary disease is by far a more common presentation compared to mammary lesions should be highlighted with more emphasis.

Lack of cohesion example - Lines 66-71

“Further, M. c. capricolum and M. putrefaciens are rarely isolated and never reported officially. Moreover, M. m. capri, M. c. capricolum and M. putrefaciens are pathogens of goats rarely affecting sheep which are by far the most economically important small ruminant species worldwide particularly in the EU where sheep numbers are seven times higher than that for goat [6].”

Or Lines 233-236

“In countries where M. agalactiae represents the most important and prevalent pathogen associated with CA, the disease shows the same clinical course both in sheep and goat  and, in those countries like Italy, where often both animal species are kept together in the same group, owners do not report any clinical differences between them.

Line 75

“As stated above, the historical and major cause of CA is M. agalactiae which was first isolated from affected sheep in 1923 in France” – does not add new information, consider removing this sentence.

Line 146

“previously identified as …” - does not add new information, previous nomenclature already mentioned in Introduction

Line 151

“ while other routes failed to produce pathology [24]” – failed to produce lesions

On another note, I would be interested to know if the authors would go a step further and propose another syndrome terminology for disease processes caused by M. m. capri or M. c. capricolum in particular. It might be a bit of a stretch at this stage as more studies around the pathogenesis of these agents are needed but I would be interested to know the authors’ view on this.

Author Response

The authors promote a worthwhile discussion regarding the aetiology of contagious agalactia. Evidences such as clinical presentation, pathogenesis and geographic distribution are discussed in an appropriate depth. The text organization is logical . However, the article would benefit from more work in connecting ideas at sentence level. Sentences are often too long which hampers the cohesion of arguments presented.  As this scientific paper style strongly relies on an effective argumentation process, careful check and proofread of the manuscript is strongly recommended.

Thanks we have carefully been through the manuscript and hope the revised edition has been improved

In addition, the fact that Mycoplasma mycoides subsp. Capri appears to have a different pathogenesis, as pulmonary disease is by far a more common presentation compared to mammary lesions should be highlighted with more emphasis.

We have increased emphasis of this mycoplasma

Lack of cohesion example - Lines 66-71

“Further, M. c. capricolum and M. putrefaciens are rarely isolated and never reported officially. Moreover, M. m. capri, M. c. capricolum and M. putrefaciens are pathogens of goats rarely affecting sheep which are by far the most economically important small ruminant species worldwide particularly in the EU where sheep numbers are seven times higher than that for goat [6].”

This section has been revised extensively L64-79

Or Lines 233-236

“In countries where M. agalactiae represents the most important and prevalent pathogen associated with CA, the disease shows the same clinical course both in sheep and goat  and, in those countries like Italy, where often both animal species are kept together in the same group, owners do not report any clinical differences between them.

Sentences changed L244-247

Line 75

“As stated above, the historical and major cause of CA is M. agalactiae which was first isolated from affected sheep in 1923 in France” – does not add new information, consider removing this sentence.

removed

Line 146

“previously identified as …” - does not add new information, previous nomenclature already mentioned in Introduction

As the mycoides cluster may be confusing to the uninitiated, we thought it would be helpful to readers to remind them that these two mycoplasmas were separate subsps until quite recently L160-161

Line 151

“ while other routes failed to produce pathology [24]” – failed to produce lesions

The sentence was modified

On another note, I would be interested to know if the authors would go a step further and propose another syndrome terminology for disease processes caused by M. m. capri or M. c. capricolum in particular. It might be a bit of a stretch at this stage as more studies around the pathogenesis of these agents are needed but I would be interested to know the authors’ view on this.

This is a very good suggestion. We referred to MAKePs (L97) as an unsatisfactory term for diseases caused by these mycoplasmas and prefer CRAS (Caprine Respiratory and Articular Syndrome) as a more informative and accurate description. We will include this in the text  L282-287. Please let us know if you want acknowledgement

Round 2

Reviewer 1 Report

I appreciate the attention to the suggested revisions. There are some typographical errors that need to be addressed, but overall the manuscript reads well.